# TNP Analogues Inhibit the Virulence Promoting IP_3-4_ Kinase Arg1 in the Fungal Pathogen *Cryptococcus neoformans*

**DOI:** 10.3390/biom12101526

**Published:** 2022-10-20

**Authors:** Desmarini Desmarini, Daniel Truong, Lorna Wilkinson-White, Chandrika Desphande, Mario Torrado, Joel P. Mackay, Jacqueline M. Matthews, Tania C. Sorrell, Sophie Lev, Philip E. Thompson, Julianne Teresa Djordjevic

**Affiliations:** 1Centre for Infectious Diseases and Microbiology, The Westmead Institute for Medical Research, Westmead, NSW 2145, Australia; 2Sydney Institute for Infectious Diseases, Faculty of Medicine and Health, University of Sydney, Sydney, NSW 2006, Australia; 3Medicinal Chemistry, Monash Institute of Pharmaceutical Sciences, Monash University, 381 Royal Parade, Parkville, VIC 3052, Australia; 4Sydney Analytical, Core Research Facilities, The University of Sydney, Sydney, NSW 2006, Australia; 5School of Life & Environmental Sciences, The University of Sydney, Sydney, NSW 2006, Australia; 6Western Sydney Local Health District, Westmead, NSW 2145, Australia

**Keywords:** inositol polyphosphate kinase, IP_3-4_K, TNP, dibenzylaminopurine, fungal pathogens, antifungal drug discovery, structure activity relationship, *Cryptococcus neoformans*, enzyme assay, surface plasmon resonance

## Abstract

New antifungals with unique modes of action are urgently needed to treat the increasing global burden of invasive fungal infections. The fungal inositol polyphosphate kinase (IPK) pathway, comprised of IPKs that convert IP_3_ to IP_8_, provides a promising new target due to its impact on multiple, critical cellular functions and, unlike in mammalian cells, its lack of redundancy. Nearly all IPKs in the fungal pathway are essential for virulence, with IP_3-4_ kinase (IP_3-4_K) the most critical. The dibenzylaminopurine compound, *N*2-(*m*-trifluorobenzylamino)-*N*6-(*p*-nitrobenzylamino)purine (TNP), is a commercially available inhibitor of mammalian IPKs. The ability of TNP to be adapted as an inhibitor of fungal IP_3-4_K has not been investigated. We purified IP_3-4_K from the human pathogens, *Cryptococcus neoformans* and *Candida albicans*, and optimised enzyme and surface plasmon resonance (SPR) assays to determine the half inhibitory concentration (IC_50_) and binding affinity (K_D_), respectively, of TNP and 38 analogues. A novel chemical route was developed to efficiently prepare TNP analogues. TNP and its analogues demonstrated inhibition of recombinant IP_3-4_K from *C. neoformans* (*Cn*Arg1) at low µM IC_50_s, but not IP_3-4_K from *C. albicans* (*Ca*Ipk2) and many analogues exhibited selectivity for *Cn*Arg1 over the human equivalent, *Hs*IPMK. Our results provide a foundation for improving potency and selectivity of the TNP series for fungal IP_3-4_K.

## 1. Introduction

Invasive fungal diseases affect over 300 million people and cause greater than 1.5 million deaths annually around the world, matching deaths from tuberculosis and exceeding those from malaria [1,2]. *Cryptococcus neoformans* and drug-resistant *Candida* species are high on a list of a soon-to-be-released fungal priority pathogens established by the WHO. *C. neoformans* is an environment yeast that initially infects the lungs but has a predilection for the central nervous system where it causes meningoencephalitis. *C. albicans* is a component of the human mycobiota and causes candidemia.

Despite the high rates of morbidity and mortality due to invasive fungal diseases, our antifungal drug armamentarium has limitations and is confined to four classes that predominantly inhibit fungal sterols (ergosterol) and/or their synthesis, and synthesis of the cell wall. Prolonged use of the polyenes, which target fungal ergosterol (e.g., Amphotericin B), can cause kidney failure [3]. The azoles, the most advanced class, inhibit fungal growth but do not kill fungi and drug resistance is a major concern [4]. Increased human exposure to azoles used in agriculture and farming, coupled with prolonged azole treatment in the clinic, are thought to be major drivers of resistance and azole cross-resistance [4,5]. The echinocandins (target cell wall synthesis) have a limited antifungal spectrum with no activity against *C. neoformans* [6,7]. Furthermore, echinocandin-resistant strains, including the multi-drug resistant *C. auris* [8], are emerging and pose a serious global health threat. Thus, there is an urgent need to develop new antifungal drug classes with novel targets.

Recent studies in *C. neoformans* and *C. albicans* have shown that the inositol polyphosphate (IP) kinase (IPK) pathway is a promising target for antifungal drug development due to its involvement in numerous critical cellular processes (reviewed in [9,10,11,12]). The fungal IPK pathway is comprised of a series of sequentially acting IPKs that convert IP_3_ to IP_8_, with IP_3-4_K the most critical for virulence and cellular function in both *C. neoformans* [13,14,15,16] and *C. albicans* [17,18,19]. IP6K (Kcs1) and its product, IP_7_, are also essential for the virulence of *C. neoformans* [15].

In contrast to fungi, the product of IP_3-4_K, IP_5_, can be generated by more than one route in human and involves 4 different enzymes: inositol polyphosphate multikinase (*Hs*IPMK), IP3K, INPP5 and ITPK1 (reviewed in [10,11]). In addition to having IP_3-4_K activity, *Hs*IPMK functions as a phosphoinositide 3-kinase [20,21,22] and as a scaffold protein in the target of rapamycin (TOR) where its enzymatic activity is not required [23] (reviewed in [24]). *Hs*IP6K has been implicated in various health conditions including cancer [25,26], aging [27] and diabetes and obesity [28].

Based on the critical role of the IPK pathway in fungal virulence and the redundancy in the human pathway, targeting fungal IP_3-4_K is a promising strategy for developing a novel class of antifungal drug. Firstly, IP_3-4_K is a different target to the current antifungal drug targets and impacts numerous cellular functions critical for virulence rather than a single function. Secondly, fungal and human IP_3-4_Ks share a low amino acid sequence homology, with commonality restricted to a few key catalytic residues [11]. Thirdly, IPK function is non-redundant in fungi, with each step catalysed by a single enzyme. This is in contrast to mammalian cells where the pathway is more branched, with at least 3 IPKs catalysing the conversion of IP_3_ to IP_5_. The latter points suggest that inhibitors specific for the fungal enzyme can be generated without detrimentally impacting the human IPK pathway. 

The search for human IPK inhibitors, especially IP6K inhibitors, to serve as metabolic regulators has been ongoing for over 20 years with several promising potent molecules produced (reviewed in [29]). The spiro-oxindole-based compounds SC-919 [30], LI-2242 and LI-2172 [31] possess nanomolar potencies against all *Hs*IP6K isoforms in vitro. The benzoisoxazole-based UNC7467 achieved selectivity for IP6K1 and IP6K2 with nanomolar potencies compared to IP6K3 [32]. Currently, *N*2-(*m*-trifluorobenzylamino)-*N*6-(*p*-nitrobenzylamino) purine (TNP), is the only commercially available IPK inhibitor (reviewed in [29]). TNP has a dibenzylaminopurine scaffold in which two benzyl rings are attached at positions *N*2 and *N*6 of the purine structure. TNP is an ATP-competitive inhibitor of all three IP6K isoforms in mammalian cells [33] and was discovered through a purine-based compound library screen against human IP3K [34]. However, TNP was later found to be more potent against IP6K, with an IC_50_ of 0.47 µM [33] compared to an IC_50_ of 10 µM against IP3K [34], and demonstrated anti-obesity and anti-diabetic effects in a mouse model of high fat diet-induced obesity [28,35]. Although TNP is a potent inhibitor of human IP6K, it does not inhibit rat IPMK at a concentration of 10 µM [36] and its inhibitor properties against human IPMK have not been determined. 

As a probe and drug, TNP has low water solubility, which negatively impacts its potency, and has off-target effects [28,35]. Using medicinal chemistry approaches, it was shown that modifications could be introduced to TNP to improve its solubility and selectivity for the different IP6K isoforms, with a preference for IP6K1 [37], providing proof-of-principle that selectivity of TNP for other IPKs in the same family, including IP_3-4_Ks, can be achieved. In a separate study, the modification of TNP was able to reduce off-target effects by reducing inhibition of the cytochrome P450 enzyme CYP3A4 [35].

TNP treatment of the non-pathogenic yeast, *Saccharomyces cerevisiae*, reduces IP_7_ levels and phenocopies an IP6K deletion mutant by causing vacuolar fragmentation [33,38], consistent with TNP targeting yeast IP6K. However, the potency of TNP as an inhibitor of fungal IPKs in vitro was not assessed. Here, we purify tag-free IP_3-4_K from *C. neoformans* (*Cn*Arg1) and *C. albicans* (*Ca*Ipk2), and *Hs*IPMK and assess the potency and selectivity of inhibition of TNP and 38 derivatives using an optimised enzyme assay and a newly developed surface plasmon resonance (SPR) assay.

## 2. Materials and Methods

### 2.1. Strains and Media

*Yeast strains*: *Cryptococcus neoformans* var grubii (strain H99) (serotype A, MATα) and *Candida albicans* (strain SC5314), used to produce IP_3-4_K cDNA, were routinely grown on YPD (1% yeast extract, 2% peptone and 2% dextrose). *Bacterial strains*: One Shot™ TOP10 chemically competent *E. coli* (Invitrogen™, Waltham, MA, USA) were used for transformation and plasmid storage (high-copy plasmid). The chemically competent BL21 (DE3) *E. coli* strain was used for protein expression. *E. coli* strains were routinely grown in LB broth (1% tryptone, 0.5% yeast extract, 0.5% NaCl) or LB Agar (same recipe as broth with 1.5% agar) with or without Ampicillin (100 µg/mL) to retain plasmids. S.O.C media (2% tryptone, 0.5% yeast extract, 0.05% NaCl, 2.5 mM KCl, 20 mM glucose, 5 mM MgCl_2_, pH 7) and LB broth were used for post-transformation cell recovery.

### 2.2. RNA Extraction and cDNA Synthesis

The mRNA sequences of Arg1 in *Cryptococcus neoformans* (NCBI Reference Sequence: XM_012198137.1) and Ipk2 in *Candida albicans* (NCBI Reference Sequence: XM_709458.2) were retrieved from the NCBI database. RNA was extracted from each strain using TRIzol™ (Invitrogen) as per the protocol described in [14,39]. Briefly, YPD overnight-grown fungal cells were pelleted by centrifugation and snap-frozen in liquid nitrogen. TRIzol™ and 425–600 µm glass beads (Sigma, St. Louis, MO, USA) were added to the cell pellets, which were homogenised by bead-beating. RNA was extracted following the manufacturer’s instructions, residual DNA was removed by RQ DNaseI treatment (Promega) and cDNA was synthesized using Moloney Murine Leukemia Virus Reverse Transcriptase (Promega).

### 2.3. Cloning of IP_3-4_K into the pGEX-6P Expression Vector

pGEX-6P_CnArg1 and pGEX-6P_*Ca*Ipk2: IP_3-4_K cDNA created above was used as a template to PCR-amplify *Cn*Arg1 using primers *ARG1-BglII-s* and *ARG-EcoRI-a* and *Ca*Ipk2 with primers *CaIPK2-BamHI-s* and *CaIPK2-XhoI-a* (see Table 1), using Invitrogen™ Platinum™ Taq High Fidelity DNA Polymerase. The PCR products were cloned into the pGEX-6P expression vector, and transformed into TOP10 competent cells. Colonies were screened by colony PCR using primers pGEX-Seq-s and pGEX-Seq-a to identify those with the correct insert. The inserts were sequenced to confirm the absence of PCR-induced mutations by comparison to the NCBI database, and that the cDNA was in-frame with the GST to ensure proper translation. 

*pGEX-6P_HsIPMK*: human IPMK (ORF NM_152230.5) cloned into pGEX-6P. This plasmid was ordered from GenScript using their Express Cloning service including sequencing by GenScript to ensure the correct sequence was provided. 

### 2.4. Expression and Purification of IP_3-4_K Proteins

pGEX-6P_*Cn*Arg1, pGEX-6P_*Ca*Ipk2 and pGEX-6P_*Hs*IPMK plasmids were used to transform chemically competent BL21 (DE3) cells. Transformed BL21 (DE3) cells were used to inoculate LB-Ampicillin broth grown overnight at 37 °C with shaking. This starter culture was used to seed fresh LB-Ampicillin medium (1:200 dilution), which was then incubated at 37 °C with shaking until the OD_600_ reached 0.6. IPTG (1 mM) was then added to induce protein expression overnight. Cells were harvested and the cell pellet was resuspended in GST lysis buffer (20 mM HEPES, pH 7.3, 100 mM NaCl, 1 mM EDTA, 1 mM EGTA, 0.5% Triton X-100, 2 mM DTT, 1 mM PMSF, Roche cOmplete™ Protease Inhibitor Mini Tablets, EDTA-free), probe-sonicated and centrifuged to remove debris. The cleared lysate (supernatant) containing soluble GST-tagged fusion protein was then collected and subjected to two rounds of purification.

The first purification step involved incubating the clear lysate with Glutathione sepharose^®^4B (GE Life Sciences, Chicago, IL, USA) beads at 4 °C with gentle end-to-end rotation. The suspension was then added to an empty column for gravity flow chromatography, and unbound proteins (flow-through) were separated from bead-bound proteins. The beads were then washed 6 times with 3 column volumes of the following ice-cold buffers each time: twice with lysis buffer, twice with wash buffer 1 (50 mM Tris-HCl, pH 7.5, 500 mM NaCl, 2 mM EDTA, 1 mM EGTA, 1% Triton X-100, 2 mM DTT) and twice with wash buffer 2 (50 mM Tris-HCl, pH 8.0, 150 mM NaCl, 2 mM DTT). For protein elution, the washed beads were incubated in elution buffer (50 mM Tris-HCl, pH 8.0, 150 mM NaCl, 2 mM DTT, 150 µg/mL GST-HRV 3C protease) at 4 °C overnight with gentle end-to-end rotation. The eluted protein solution was then collected, and the beads were further washed with wash buffer 2 to maximise protein recovery. All the eluted protein was pooled and concentrated using an Amicon^®^ Ultra-15 mL Centrifugal Filter Unit (Merck Millipore, Burlington, MA, USA) with a 10 kDa molecular weight cut-off (MWCO). 

The second purification step involved size exclusion chromatography (SEC) using an AKTA Fast Protein Liquid Chromatography (FPLC) system (GE Life Sciences, Chicago, IL, USA) with a HiLoad 26/600 Superdex 75 pg column (GE Life Sciences, Chicago, IL, USA). Fractions were collected, pooled and concentrated using an Amicon^®^ Ultra-15 mL Centrifugal Filter Unit (Merck Millipore, Burlington, MA, USA) with a 10 kDa MWCO. Enzyme purity and size were assessed by SDS-PAGE (4–12% Bis-Tris protein gel). The protein multimerization state and a more accurate molecular weight were determined using SEC-MALLS (size exclusion chromatography coupled to multi angle laser light scattering (Wyatt Technology, Santa Barbara, CA, USA)). 

### 2.5. Determination of K_m_ and V_max_ for ATP

The kinetic properties of the purified recombinant enzymes were determined and compared using a Kinase-Glo^®^ Max Luminescent Kinase Assay kit (Promega, Madison, WI, USA) in a reaction buffer consisting of 20 mM HEPES pH 6.8, 100 mM NaCl, 6 mM MgCl_2_, 20 µg/mL BSA, 1 mM DTT [14]. The IP_3_ concentration was fixed at 200 µM. The ATP starting concentrations ranged between 25 µM to 400 µM and the ATP remaining after up to 10-min reaction time was assessed by adding Kinase-Glo^®^ reagent. An ATP standard curve (0–500 µM) was also generated using the same kit. The ATP concentration remaining was measured as luminescence using a SpectraMax iD5 plate reader. The relative luminescence unit (RLU) was converted to ATP concentration using the ATP standard curve. The reaction velocity was calculated for each starting ATP concentration and the data was fitted to the Michaelis-Menten equation using GraphPad Prism 9 to obtain V_max_ and K_m_ values.

### 2.6. Enzyme Activity and Inhibition Assay to Screen for ATP-Competitive Inhibitors

The 2,6-disubstituted purine compounds are ATP-competitive IPK inhibitors. The screening assay was therefore set up to favour screening of ATP-competitive inhibitors following the manufacturer’s suggestion. This involved using 10 µM ATP and a Kinase-Glo kit that measures up to 10 µM ATP (Kinase-Glo^®^ Luminescent Kinase Assays, Promega, Madison, WI, USA). The amount of IP_3_ and kinase used for each recombinant enzyme was determined following the manufacturer’s instructions, and this concentration was used to assess the inhibitory properties of TNP analogues against each of the recombinant enzymes. 

The assay was carried out in reaction buffer (20 mM HEPES pH 6.8, 100 mM NaCl, 6 mM MgCl_2_, 20 µg/mL BSA, 1 mM DTT) containing 10 µM ATP and the optimal amount of IP_3_ and enzyme determined via the optimisation process in a final volume of 50 µL. The analogues were dissolved in DMSO. For the assay, the inhibitors were added to the reaction mixture to achieve a final concentration of 50 µM in 5% DMSO (1:20 dilution). To achieve inhibitor concentrations lower than 50 µM in the assay, the inhibitor stock was first serially diluted two-fold in DMSO and added to the reaction mixture, to achieve concentrations ranging between 0.8 μM and 50 µM in 5% DMSO. Once all analogues and their dilution series had been added, the reaction was started by adding an optimal amount of recombinant IP_3-4_K enzyme. The reaction was stopped after 10-min incubation at room temperature by adding a 50 µL of Kinase-Glo^®^ reagent. The mixture was incubated at room temperature in the dark for 10–15 min. Luminescence (RLU) was measured using a SpectraMax iD5 and an integration time of 0.5 s. Percent enzyme activity was defined as (RLU_negative_ − RLU_sample_)/(RLU_negative_ − RLU_positive_) × 100. ‘Positive’ refers to 100% enzyme activity (no inhibitor used as the positive control), while ‘negative’ refers to 0% enzyme activity (no enzyme used as the negative control). The IC_50_ was calculated using GraphPad Prism 9 by plotting the RLU, which indicates the ATP concentration remaining, against the concentration of TNP analogue. 

### 2.7. Surface Plasmon Resonance (SPR) to Assess Binding Affinity of TNP Analogues

SPR was carried out on a Biacore T200 (Cytiva, Marlborough, MA, USA). A streptavidin surface was prepared by amine coupling streptavidin to a CM5 chip using standard procedures in 20 mM HEPES, 150 mM NaCl pH 7.5 at 37 °C. Briefly, the surface was activated by injection of 1:1 NHS:EDC (*N*-ethyl-*N*’-(3-(dimethylamino)propyl)carbodiimide/*N*-hydroxysuccinimide) followed by a 7-min injection of 100 µg/mL streptavidin in 10 mM sodium acetate (pH 4.5) at a flow rate of 2 µL/min. Unreacted groups on the surface were blocked by injection of 1 M ethanolamine (pH 8.0). Purified recombinant *Cn*Arg1 was N-terminally biotinylated and immobilised to a level of ~4000 RU onto the streptavidin surface at 25 °C in 20 mM HEPES, 200 mM NaCl, 5 mM MgCl_2_ (pH 7.5) at a flow rate of 2 µL/min.

For analysis, compounds were prepared to 50 mM in 100% DMSO and diluted to the desired concentration in SPR running buffer; 20 mM HEPES (pH 7.5), 200 mM NaCl, 5 mM MgCl_2_, 5% DMSO. Analysis was carried out using multi-cycle kinetics over a compound concentration range of 6.25–200 µM in running buffer at 10 °C, with an association time of 60 s and a dissociation time of 120 s. Data were reference subtracted, and a solvent correction applied. Analysis was carried out using Biacore T200 Evaluation Software and all data fit to a 1:1 Langmuir binding isotherm. 

### 2.8. Synthesis of 2,6-Disubstituted Purine Analogues

#### 2.8.1. General Experimental Procedure

All reagents were purchased and used without further purification. Silica gel was used for column chromatography purification. ^1^H NMR and ^13^C NMR spectra were collected on a Bruker Advance III Nanobay 400 MHz spectrometer (^1^H at 400 MHz and ^13^C at 100 MHz). All spectra were processed using MestReNova 11.0 software. The chemical shifts of ^1^H and ^13^C are reported in parts per million (ppm) and were measured relative to the expected chemical shifts of the NMR solvents; CDCl_3_, 7.26 (77.16 for ^13^C NMR) CD_3_OD, 3.31 (49.00 for ^13^C NMR) and DMSO-*d*_6_, 2.50 (39.52 for ^13^C NMR). The format used to report the spectra was as follows: chemical shift (multiplicity, coupling constant (if applicable), integration). Multiplicity was defined as: s = singlet, d = doublet, t = triplet, q = quartet, sd = singlet of doublets, dd = doublet of doublets, dt = doublet of triplets, tt = triplet of triplets and m = multiplet. Apparent splitting was abbreviated as app. and a broad resonance was abbreviated as br. Coupling constants were reported as *J* in Hertz (Hz). 

All analytical HPLC analyses were done on an Agilent 1260 Infinity Analytical HPLC coupled with a 1260 Degasser: G1322A, 1260 Binary Pump: G1312B, 1260 HiP ALS autosampler: G1367E, 1260 TCC: G1316A and 1260 DAD detector: G4212B. The column used was a Zorbax Eclipse Plus C18 Rapid Resolution 4.6 × 100 mm 3.5-micron. The sample injection volume was 2 μL which was run in 0.1% TFA in acetonitrile at a gradient of 5–100% over 10 min with a flow rate of 1 mL/min. Detection methods were with 214 nm and 254 nm.

All HRMS analyses were done on an Agilent 6224 TOF LC/MS Mass Spectrometer coupled to an Agilent 1290 Infinity (Agilent, Palo Alto, CA, USA). All data were acquired, and reference mass corrected via a dual-spray electrospray ionization (ESI) source. Each scan or data point on the Total Ion Chromatogram (TIC) is an average of 13,700 transients, producing a spectrum every second. Mass spectra were created by averaging the scans across each peak and background subtracted against the first 10 s of the TIC. Acquisition was performed using the Agilent Mass Hunter Fata Acquisition software version B.05.00 Build 5.0.5042.2 and analysis was performed using Mass Hunter Qualitative Analysis version B.05.00 Build 5.0.519.13.

#### 2.8.2. General Procedure I for the First Nucleophilic Aromatic Substitution in the 6-Position (2a and 2b)

To a solution of 6-chloro-2-fluoropurine (1 mmol, 1.00 equiv.) in *n*-butanol (4 mL), was added *N*,*N*-diisopropylethylamine (1.41 equiv.). The mixture was stirred at room temperature for 5 min. To the mixture, was added an amine (1.02 equiv.). The mixture was warmed to 65 °C and stirred for 4 h. Then, the reaction mixture was concentrated *in vacuo* and to the residue, was added cold water. The precipitate was filtered, washed with cold water and purified.

#### 2.8.3. General Procedure II for the Second Nucleophilic Aromatic Substitution in the 2-Position (9 to 29 (Excluding 14 and 17), 32, 35, 38 and 39)

To a solution of a *N*6-substituted-2-fluoro-9*H*-purin-6-amine (1 mmol, 1.00 equiv.) in *n*-butanol (4 mL), was added *N*,*N*-diisopropylethylamine (2.20 equiv.). The mixture was stirred at room temperature for 5 min. To the mixture, was added an amine (2.00 equiv.). The mixture was heated to reflux and stirred overnight (16 h). Then, the reaction mixture was concentrated *in*
*vacuo* and purified.

#### 2.8.4. General Procedure III for the Reductive Amination in the 2-Position (5a to 5c)

To a solution of 2-amino-6-chloropurine (1 mmol, 1.00 equiv.) in ethyl acetate (4 mL), was added a benzaldehyde (1.20 equiv.). The mixture was cooled to 0 °C and was added trifluoroacetic acid (2.24 equiv.) and sodium triacetoxyborohydride (1.50 equiv.). The reaction was warmed to room temperature and stirred overnight (16 h). The mixture was quenched with 10% aq. NaOH solution (20 mL) to pH ~8–9 and extracted with ethyl acetate (3 × 20 mL). The organic extracts were combined, washed with brine (50 mL) and dried over MgSO_4_. Then, it was concentrated *in vacuo* and purified.

#### 2.8.5. General Procedure IV for the Nucleophilic Aromatic Substitution in the 6-Position (14, 30, 31, 36 and 41 to 44)

To a solution of *N*2-substituted-6-chloro-9*H*-purin-2-amine (1 mmol, 1.00 equiv.) in *n*-butanol (4 mL), was added an amine (1.50 equiv.) and triethylamine (1.00 equiv.). The reaction was heated to reflux and stirred for 5 h. Then, the reaction mixture was concentrated *in vacuo* and purified.

Additional synthetic methods and spectroscopic data for individual compounds is provided in the Appendix A.

## 3. Results

### 3.1. Purification of IP_3-4_K from C. neoformans, C. albicans and Human

IP_3-4_K from *C. neoformans* (*Cn*Arg1) and *C. albicans* (*Ca*Ipk2) and *Hs*IPMK were expressed from pGEX-6P in *E. coli* as GST fusion proteins and purified in a two-step process involving glutathione-affinity chromatography, followed by GST cleavage by HRV 3C protease and size exclusion chromatography. SDS-PAGE indicated *Cn*Arg1 and *Ca*Ipk2 have molecular weights close to the predicted values of 49 kDa and 40 kDa, respectively, and a purity of >95% (Figure 1A). SEC-MALLS was used to assess the oligomeric state and was consistent with both IP_3-4_K enzymes being monomeric. The solution molecular weights of *Cn*Arg1 and *Ca*Ipk2 were calculated as 52 kDa and 43 kDa, respectively (i.e., within 10% of the predicted monomer molecular weight) (Figure 1B). From SDS-PAGE, the molecular weight of *Hs*IPMK was ~47 kDa (Figure 1A), in agreement with the predicted molecular weight of 47.2 kDa [40]. *Hs*IPMK purity was ~60%. Reduced *Hs*IPMK purity could be due to limited expression levels and contamination with bacterial heat shock proteins. *Hs*IPMK was previously shown to be monomeric [21,22,40,41]. The purification yielded ~3.6 mg per litre of *Cn*Arg1, ~8.8 mg per litre of *Ca*Ipk2 and ~0.3 mg per litre of *Hs*IPMK proteins.

### 3.2. Establishing the Kinetic Properties of IP_3-4_K

A luminescence assay was optimised from [14] to determine the kinetic properties of *Cn*Arg1 and *Ca*Ipk2 using IP_3_ as the substrate. For *Cn*Arg1 and *Ca*Ipk2, the K_m_ was 300 ± 67 µM and 213 ± 49 µM, and the V_max_ was 12 ± 1.5 and 21 ± 3 µmol ATP/mg protein/min, respectively (Figure 2). Using the same assay conditions, *Hs*IPMK had a lower K_m_ (128 ± 35 µM) compared to the fungal IP_3-4_K enzymes and a lower V_max_ (0.6 ± 0.07 µmol ATP/mg protein/min) (Figure 2), consistent with a higher affinity for ATP. This compares to a previously determined K_m_ for ATP of 61 ± 6 µM when PIP_2_ was used as the substrate [22] and 10 µM when IP_4_ was used as the substrate [31]. The K_m_ of rat IPMK for ATP was previously determined to be 64 µM when IP_3_ was used as the substrate [36].

### 3.3. Optimizing the Enzyme Assay and Determining the IC_50_ of TNP

Next, we determined the inhibitory properties of TNP. TNP is an ATP-competitive inhibitor [34]. To bias the assay toward ATP-competitive inhibitors, the ATP concentration was reduced from 500 µM to 10 µM, which is a value lower than the K_m_ of all IP_3-4_K enzymes (Figure 2). IP_3_ and enzyme concentrations were also optimised following the manufacturer’s instructions. Briefly, this involved measuring ATP consumption using a fixed saturated amount of kinase and varying the concentration of IP_3_ from 0 to 200 µM. The IP_3_ concentration that resulted in the largest change in luminescence was chosen as the optimal IP_3_ concentration. This optimal amount of IP_3_ was then used to test varying concentrations of the kinase ranging from 0 to 20 ng/µL. The amount of kinase that produced luminescence values in the linear range of the kinase titration curve was deemed to be the optimal kinase amount. The optimised concentrations of enzyme assay components are summarised in Table 2.

Using the optimized assay conditions (Table 2), TNP was found to inhibit *Cn*Arg1 and *Hs*IPMK with an IC_50_ of 21 ± 6 µM and 7 ± 0.5 µM, respectively (Figure 3). To our knowledge, this is the first report that TNP inhibits *Hs*IPMK. Interestingly, TNP did not inhibit *Ca*Ipk2.

### 3.4. Synthesis of 2,6-Disubstituted Purine Analogues 

TNP analogues were synthesized using multiple strategies that expedited the inclusion of substituents at the *N*2- and *N*6-positions of the purine scaffold (Table 3, Table 4 and Table 5). In addition, modification at *N*9 was included in examples 45 and 46 (Table 6). Note that several of these compounds were previously reported in studies of human IPKs as well as other enzyme targets [34,35,37,42,43].

For fixed *N*6-amine substitutions, 2-fluoro-6-chloro-9*H*-purine, **1** was subjected to consecutive nucleophilic aromatic substitutions (Figure 1) [37,44,45]. While substitution of **1** occurs in the 6-position preferentially, giving **2** [46], bis-substitution was a common observation initially. Monitoring the temperature and length of the first nucleophilic aromatic substitution reaction gave improved outcomes. The second substitution at the 2-position yielded the target compounds, **3** but also proved to be challenging, with reactions requiring high reagent concentrations to proceed. 

In order to diversify the 6-position conveniently, an alternate synthetic route was adopted, where the group on the 2-position was introduced first. This has been achieved previously through the use of protecting groups or modification of the 6-chloro group to bypass the more reactive groups [28,37]. However, we pursued a more direct route by a reductive alkylation of 2-amino-6-chloro-9*H*-purine (Figure 2). This was then followed by the nucleophilic aromatic substitution by various amines at the 6-chloro position to give **5**. Reductive alkylation in the 2-position has previously been demonstrated [47,48], but to our knowledge, reductive alkylation in the 6-position has not been reported before. Substitution of **5** with various amines gave target compounds **6.**

Lastly, the impact of an alkyl group in the 9-position was explored. This was accomplished through alkylation of **7** using an alkyl halide to give **8** (Figure 3). 

### 3.5. Assessment of the Inhibitory Properties of 2,6-Disubstituted Purine Analogues

Using the assay conditions established for TNP, all analogues were assessed for inhibition of *Cn*Arg1, *Hs*IPMK and *Ca*Ipk2. We found that most analogues inhibited *Cn*Arg1 with IC_50_ values ranging from 10–100 μM (summarised in Table 3, Table 4, Table 5 and Table 6), but none inhibited *Ca*Ipk2 (see Appendix A). Compound structures and their IC_50_ values against *Cn*Arg1 and *Hs*IPMK are summarised in Table 3, Table 4, Table 5 and Table 6. Only a selected number of analogues were tested for inhibitory activity against *Hs*IPMK, with a focus on those with similar or better potencies than TNP (i.e., those with an IC_50_ less than 40 µM). Note that the reduced potencies of some of these compounds could be due to poor solubility in water although this was not investigated further. Enzyme inhibition assays could not be performed on **11**, **18** and **19** as they were poorly soluble in 5% DMSO.

We first identified that **9**, which lacked the nitro group of TNP, had a similar potency to TNP against *Cn*Arg1, but slightly reduced potency against *Hs*IPMK. **9** had previously been described by [34,37] during their studies around IP3K and IP6K, respectively. They showed that the removal of the nitro group resulted in similar inhibition as TNP. Note that some structure activity relationship (SAR) against other human IPKs, IP3K and IP6K has been described before [34].

21 TNP analogues were created with modifications at the *N*2-position (see Table 3) encompassing a range of functional groups such as alternately substituted benzylamines (**10**–**24**), alternate aromatic moieties (**25**, **26**), alkyl ethers and alcohols (**27**–**29**). In some cases, activity was abolished but overall, the inhibitory activity versus *Cn*Arg1 was only modestly affected compared to **9** (<2-fold difference). However, the selectivity of compounds for *Cn*Arg1 over *Hs*IPMK was influenced by substitution. For example, **13** is a comparable inhibitor of *Cn*Arg1 to **9** but a much weaker inhibitor of *Hs*IPMK. 

Midway through this study, a research team at Johns Hopkins University published their strategy of first improving TNP solubility by replacing the *N*6-nitrobenzyl group with a methoxyethyl chain, before making further chemical modifications [37]. We employed the same strategy to explore the *N*2-position of the purine core, synthesizing a total of 11 analogues with the methoxyethyl chain at the *N*6-position (Table 4). It was reported that **30** had improved IP6K inhibition compared to TNP [37] but this was not observed with either *Cn*Arg1 or *Hs*IPMK. Swapping benzylamine substitution at *N*2 gave a range of outcomes. Arylamine substitution as **37** and **40** was tolerated (IC_50_ of 17 ± 2 µM and 14 ± 0.5 µM, respectively). 

Further variations at the *N*6-position were also considered (Table 5). One of these, **42**, which had a 4-methyltetrahydropyran group, produced the most potent inhibition against *Cn*Arg1 with an IC_50_ of 10 ± 0.5 µM. Substitution in the 9-position was not tolerated in the two analogues, **45** and **46**, which did not have any inhibitory activity for *Cn*Arg1 (Table 6).

### 3.6. Assessing Binding Affinities of TNP and Its Analogues

To complement the enzyme inhibition data, the binding affinities (K_D_) of TNP analogues were determined for *Cn*Arg1 using surface plasmon resonance (SPR) and the results are summarised in Table 7 (see Appendix A for representative SPR sensorgrams and dose–response curves). A K_D_ could not be determined for TNP and its closest analogue **9**, due to poor behaviour on the SPR chip, which is most likely attributable to poor aqueous solubility. Chemical modification of TNP and **9** improved compound properties, allowing the binding affinities of 16 analogues to be determined. Apart from **37**, these analogues bound to *Cn*Arg1, confirming a correlation between binding and inhibition. With the exception of **39** and **40**, the analogues generally had comparable IC_50_ and K_D_ values.

### 3.7. Comparing the Selectivity of TNP and Its Analogues for CnArg1

Using the IC_50_ data for TNP and the seventeen TNP analogues with the lowest IC_50_ against *Cn*Arg1, the *Hs*IPMK/*Cn*Arg1 IC_50_ ratios were calculated (see last column in Table 3, Table 4, Table 5 and Table 6) and plotted (Figure 4). Fourteen TNP analogues were found to be more selective for *Cn*Arg1 (ratio higher than 1). While unable to identify significantly more potent analogues of *Cn*Arg1 than TNP, the collected assessment of enzyme inhibitory activities did show that the relative selectivity over the human orthologue could be influenced by altered substitutions, with the selectivity ratios changing 20-fold in going from TNP (3-CF_3_ substitution) to **13** (3-OCH_3_ substitution) (Table 3).

## 4. Discussion

IP_3-4_K is critical for the virulence of *C. neoformans* and *C. albicans*. An IP_3-4_K (*Cn*Arg1) deletion mutant failed to either grow at 37 °C or establish an infection in a mouse model [14]. The mutant was also defective in producing a plethora of virulence-related phenotypes including capsule, melanin and phospholipase B, was more readily phagocytosed by blood-derived monocytes, had a cell wall defect and reduced metabolic flexibility and could not upregulate phosphate acquisition machinery in response to phosphate deprivation [13,14,15,16,49]. Attempts to delete both alleles of IP_3-4_ kinase (*Ca*Ipk2) in the diploid yeast, *C. albicans,* were unsuccessful, suggesting that IP_3-4_K is an essential protein. Using a knockdown approach, *Ca*Ipk2 was shown to be essential for hyphal development, secretion of degradative enzymes and survival inside macrophages [17]. Targeting fungal IP_3-4_K is therefore a promising strategy for developing a novel class of antifungal drug. Fungal IP_3-4_K is also an attractive drug target due to the redundancy that exists in the analogous IP_3_ to IP_5_ conversion step in the human pathway. 

Our SAR studies revealed that most TNP analogues inhibited *Cn*Arg1, but not *Ca*Ipk2, and that many TNP analogues were more selective for *Cn*Arg1 over *Hs*IPMK (summarised in Figure 4). This suggests that potential differences exist in the active site of each kinase. Like many IPK enzymes, the protein structures of *Cn*Arg1 and *Ca*Ipk2 have not been experimentally determined. To investigate possible structural differences among the three IP_3-4_K proteins that could explain our results, we utilized the AlphaFold-predicted structures for *Cn*Arg1 (UniProt: J9W3G0) and *Ca*Ipk2 (UniProt: Q59YE9) [50,51] and compared them to the published *Hs*IPMK crystal structure consisting of amino acid residues 50 to 416 in a complex with ADP (PDB: 5W2H) [21,22]. The *Hs*IPMK protein used to obtain the crystal structure had residues 263–377 deleted, which included a nuclear localisation signal.

Both the fungal IP_3-4_ kinases share only a ~20% sequence homology with *Hs*IPMK. However, an overlay of the AlphaFold models of *Cn*Arg1 and *Ca*Ipk2 with *Hs*IPMK (PDB: 5W2H) revealed a conserved catalytic core consisting of the three characteristic domains: an N-terminal α + β domain (N-lobe), a C-terminal α + β domain (C-lobe) and an inositol binding domain [21,22]. The N- and C-terminal domains each contain one anti-parallel beta sheet where ADP is sandwiched in between, making up the ATP binding site (Figure 5A,B). Seven out of twelve of the amino acids in the *Hs*IPMK active site that interact with ATP are conserved in *Cn*Arg1 and *Ca*Ipk2 (Figure 5C,D) and represent Pro^111^, Leu^254^, Asp^144^, Ile^384^, Asp^385^ and Lys^75^ (Figure 5E). In *Hs*IPMK, Leu^254^ and Ile^384^ form van der Waals interactions with each other and Asp^144^ forms a hydrogen bond with the ribose group of ADP. Asp^385^ interacts with the two magnesium ions, which in turn contact the two phosphate groups of ADP [21]. Ile^384^ and Asp^385^ are part of an Ile-Asp-Phe tripeptide, which is conserved in the IPK family and known to interact with a metal cofactor [11,21,52]. Lys^75^ forms a salt bridge with the alpha-phosphate of ADP [21]. 

Differences between the fungal IP_3-4_Ks and *Hs*IPMK (Figure 5E, circled residues) are the replacement of the negatively charged Asp^132^ in *Hs*IPMK with the polar asparagine in *Cn*Arg1 (Asn^100^) and *Ca*Ipk2 (Asn^102^) and replacement of Glu^131^ in *Hs*IPMK with Ala^99^ in *Cn*Arg1 and Ser^101^ in *Ca*Ipk2. In *Hs*IPMK, Glu^131^ and hydrophobic Val^133^ form hydrogen bonds with the *N*6 and *N*1 atoms of adenine, respectively. Residues within the ATP binding site that are unique to *Ca*Ipk2 are Phe^28^, which is replaced by valine in *Hs*IPMK (Val^73^) and *Cn*Arg1 (Val^32^). In *Hs*IPMK, Val^73^ makes a van der Waals interaction with the adenine group of ADP [21]. The aromatic group on Phe^28^ could potentially interfere with the binding of TNP and its analogues. Other differences are Ser^103^ in *Ca*Ipk2 replacing Val^133^ (in *Hs*IPMK) and Leu^101^ (in *Cn*Arg1), and Cys^21^ in *Ca*Ipk2 replacing Ile^65^ (in *Hs*IPMK) and Val^23^ (in *Cn*Arg1). Ile^65^ in *Hs*IPMK makes van der Waals interaction with the adenine group of ADP.

Human IP6K2 was modelled using the *Entamoeba histolytica* (*Eh*) IP6KA crystal structure (PDB ID: 4O4D) and was subsequently used for SAR analysis with flavonoids [41] and TNP [35]. Similarly, differences in the fungal IP_3-4_K active site residues can be further explored through molecular docking and/or via site-directed mutagenesis to identify residues responsible for the lack of inhibition observed for *Ca*Ipk2 and to facilitate the design of a more potent inhibitor of *Cn*Arg1, which also inhibits *Ca*Ipk2. This, TNP derivatives may become relevant to *Ca*Ipk2 inhibition once the affinity of the compounds is increased to give IC_50_ in the nanomolar range. 

## 5. Conclusions

In summary, we have demonstrated that TNP and a series of analogues derived from it, inhibit purified, tag-free IP_3-4_K produced by the human fungal pathogen, *C. neoformans*. We also show that the relative selectivity of TNP and its analogues over the human orthologue can be influenced 20-fold by making substitutions at the *N*2 position of the purine. Recently released AlphaFold protein models of fungal pathogen IP_3-4_Ks have also revealed amino acid differences in the human and fungal ATP binding site, which are being used in combination with the assays developed to guide ongoing SAR studies aimed at improving the potency and selectivity of the TNP analogues for the fungal enzyme targets, and testing new IPK inhibitor scaffolds.

## Data Availability

Not applicable.

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
