# Peer review of "TNP Analogues Inhibit the Virulence Promoting IP3-4 Kinase Arg1 in the Fungal Pathogen Cryptococcus neoformans"

_biomolecules, 2022, doi:10.3390/biom12101526_

Round 1
Reviewer 1 Report
The manuscript by Desmarini et al. reports the synthesis of 38 TNP analogues and evaluated them targeting to fungal IP3-4K. The synthetic route was reported as new with higher efficiency. IC50 values were determined by conventional kinase-glo assay and Kd values were determined by SPR assays. For comparison, human equivalent IPMK was also tested. Although the data has limited positive outcomes and discussion was made, the manuscript is scientifically designed and presented. I have the following concerns:
1, A comparison of inositol phosphate signaling pathway between fungal and mammalian cells can be included for better readability.
2, Latest development of inositol phosphate inhibitors should be included in the introduction.
3, Thanks for the details of the protein preparation. What is the yield of purification?
4, Km values can be shown in Figure2. I think N should be greater than 3 for better accuracy, especially for the determination of kinetic parameters.
5, TNP was reported and widely used as a human IP6K inhibitor. It is reported as an ATP competitive inhibitor. However, since the IC50 values for both CnArg1 and HsIPMK are significantly lower than that of human IP6K, is a non-specific inhibition possible? I think an ATP dose should be tested for TNP. Another reason I asked for this is that in Figure 3, the curve did not go to zero. Any comments? Due to the low solubility of TNP?
6, A representative of surface plasmon resonance raw data can be included.
7, The presentation of Figure 5 needs to be improved. For panels A and B, domains should be colored and labeled separately (N-lobe, C-lobe). The hinge region should be highlighted. Not sure why including yellow dashed lines over here. Don’t think valence bonds are a good choice over here since there is no distinction for P-O bonds just based on crystal structures. For panels C and D, the better way is to do ConSurf analysis (ConSurf 2016: an improved methodology to estimate and visualize evolutionary conservation in macromolecules - PMC (nih.gov)). For panel E, again, are valence bonds necessary?
Author Response
The manuscript by Desmarini et al. reports the synthesis of 38 TNP analogues and evaluated them targeting to fungal IP3-4K. The synthetic route was reported as new with higher efficiency. IC50 values were determined by conventional kinase-glo assay and Kd values were determined by SPR assays. For comparison, human equivalent IPMK was also tested. Although the data has limited positive outcomes and discussion was made, the manuscript is scientifically designed and presented. I have the following concerns:
1, A comparison of inositol phosphate signaling pathway between fungal and mammalian cells can be included for better readability.
Response: The requested information has been added to the introduction, reflected in addition to line 67-76.
2, Latest development of inositol phosphate inhibitors should be included in the introduction.
Response: The requested information has been added to the introduction, reflected in addition to line 88-93, and 109-110.
3, Thanks for the details of the protein preparation. What is the yield of purification?
Response: the protein yields have been added to the purification section of the results (section 3.1, line 320-322).
4, Km values can be shown in Figure2. I think N should be greater than 3 for better accuracy, especially for the determination of kinetic parameters.
Response: For the fungal kinetic curves, the duplicate values at each ATP concentration are consistent and justify the use of duplicates. As pointed out by the reviewer, there is variability for the human enzyme, even with n = 4, which is most extensive for the 500 µM concentration. This variability is most likely due to the low rate of enzyme activity compared to the fungal enzymes and because the results may be artefactual at such a high concentration of ATP. As further increasing the number of replicates is unlikely to eradicate the variability, we have removed the 500 µM point from each of the curves. The Km values are now shown in Figure 2. As a result of removing the 500 µM data points, the Km and Vmax values have slightly changed, and this is now reflected in the text (from line 336 to 340).
5, TNP was reported and widely used as a human IP6K inhibitor. It is reported as an ATP competitive inhibitor. However, since the IC50 values for both CnArg1 and HsIPMK are significantly lower than that of human IP6K, is a non-specific inhibition possible? I think an ATP dose should be tested for TNP. Another reason I asked for this is that in Figure 3, the curve did not go to zero. Any comments? Due to the low solubility of TNP?
Response: The reviewer is correct. Poor solubility is why the curve doesn’t go down to zero. We initially performed the inhibition assays using a higher concentration of ATP (500 µM). However, consistent with TNP being an ATP-competitive inhibitor, we could not obtain reliable IC50 curves at the higher ATP concentrations and it was difficult to read an IC50 within the solubility limits of TNP.
6, A representative of surface plasmon resonance raw data can be included.
Response: The requested data has been included as supplemental data Figure S3. A sentence was also added to line 465-466: “(see Figure S3 for representative SPR sensorgrams and dose-response curves).
7, The presentation of Figure 5 needs to be improved. For panels A and B, domains should be colored and labeled separately (N-lobe, C-lobe). The hinge region should be highlighted. Not sure why including yellow dashed lines over here. Don’t think valence bonds are a good choice over here since there is no distinction for P-O bonds just based on crystal structures. For panel E, again, are valence bonds necessary?
Response: We have now indicated the locations of the N- and C-lobe and hinge regions in the diagram following the reviewer’s suggestion. The yellow dashed lines have also been removed in Panels A and B as the interacting sidechains were omitted for clarity. However, we felt it was useful to retain these valence bonds in Panel E as the interacting side chains are indicated. The figure legend has also been modified to reflect this change (lines 508 to 518).
For panels C and D, the better way is to do ConSurf analysis (ConSurf 2016: an improved methodology to estimate and visualize evolutionary conservation in macromolecules - PMC (nih.gov)).
Response: We thank the reviewer for their suggestion and introduction to ConSurf. However, the panels C and D are aimed at highlighting only the specific differences between the ATP binding pocket residues in HsIPMK from CnArg1 and CaIpk2, the three proteins which are the focus of this study. A ConSurf depiction would involve a broader analysis of larger sequence sets, delving deeper into evolutionary conservation which is beyond the scope of this project.
Reviewer 2 Report
Fungal pathogens are a serious threat to human health and the present date antifungal-drugs are insufficient to combat serious infections. Thus, the development of new antifungal drugs is of importance. In this ms, the authors propose the development of a new drug that targets fungal inositol polyphosphate kinases IP3-4K but not the human homologue. To this end TNP (N2-(m-trifluorobenzyl), N6-(p-nitrobenzyl)purine), an established pan-IP6-Kinase inhibitor that functions in an ATP competitive manner and TNP analogues were used to determine their inhibitory effect on fungal and the human IP3-4K in vitro.
The experiments are clearly explained and well conducted. However, my concern is that the experiments show the opposite of what the authors have described as their goal.
1. Briefly, using recombinant purified IP3-4K enzymes from C. albicans (CaIpk2), C. neoformans (CnArg1) and human HsIPMK, the authors assessed inhibition by TNT. While HsIPMK and CnArg1 enzymatic activity was inhibited by TNP, CaIpk2 was not. This is not mentioned in the abstract, although this is an important finding as it suggests that the data obtained are relevant for the C. neoformans member of the IP3-4K family but not for other fungal family members. In particular, although possible differences between CaIpk2 and CnArg1 that might have led to this result are discussed in the discussion- other fungal IP3-4K have not been included. Which protein is the odd man out?
2. The authors show differences between the ability of their TNP analogues to inhibit CnArg1 versus HsIPMK. However, TNP is an established pan-IP6-Kinase inhibitor and thus developing such an inhibitor (or analogue) as an antifungal drug requires the demonstration that human inositol polyphosphate kinases are not affected. How the authors would achieve this, is not discussed.
Author Response
Fungal pathogens are a serious threat to human health and the present date antifungal-drugs are insufficient to combat serious infections. Thus, the development of new antifungal drugs is of importance. In this ms, the authors propose the development of a new drug that targets fungal inositol polyphosphate kinases IP3-4K but not the human homologue. To this end TNP (N2-(m-trifluorobenzyl), N6-(p-nitrobenzyl)purine), an established pan-IP6-Kinase inhibitor that functions in an ATP competitive manner and TNP analogues were used to determine their inhibitory effect on fungal and the human IP3-4K in vitro.The experiments are clearly explained and well conducted.
However, my concern is that the experiments show the opposite of what the authors have described as their goal.
Response: The goal of the study as mentioned in the abstract was to investigate the ability of TNP to be adapted as a selective inhibitor of fungal IP3-4K. We have now changed this to state that the goal of the study is to investigate the ability of TNP to inhibit fungal IP3-4K.
- Briefly, using recombinant purified IP3-4K enzymes from albicans(CaIpk2), C. neoformans (CnArg1) and human HsIPMK, the authors assessed inhibition by TNT. While HsIPMK and CnArg1 enzymatic activity was inhibited by TNP, CaIpk2 was not. This is not mentioned in the abstract, although this is an important finding as it suggests that the data obtained are relevant for the C. neoformans member of the IP3-4K family but not for other fungal family members. In particular, although possible differences between CaIpk2 and CnArg1 that might have led to this result are discussed in the discussion- other fungal IP3-4K have not been included. Which protein is the odd man out?
Response: We have added the CaIpk2 results to the abstract (line 33-34). We agree that the data obtained are relevant for C. neoformans IP3-4K. However, it may be too early to conclude that TNP derivatives are only relevant for inhibiting C. neoformans IP3-4K. Inhibition of the Candida enzyme may become relevant once the affinity of the compounds is increased to give IC50 in the nanomolar range. We have therefore modified the following sentence in the discussion
(lines 558-563)
Similarly, differences in the fungal IP3-4K active site residues can be further explored through molecular docking and/or via site-directed mutagenesis to facilitate the design of more potent and pan-fungal IP3-4K inhibitors.
To
Similarly, differences in the fungal IP3-4K active site residues can be further explored through molecular docking and/or via site-directed mutagenesis to identify residues responsible for the lack of inhibition observed for CaIpk2 and to facilitate the design of a more potent inhibitor of CnArg1 which also inhibits CaIpk2. Thus, TNP derivatives may become relevant to CaIpk2 inhibition once the affinity of the compounds is increased to give IC50 in the nanomolar range.
- The authors show differences between the ability of their TNP analogues to inhibit CnArg1 versus HsIPMK. However, TNP is an established pan-IP6-Kinase inhibitor and thus developing such an inhibitor (or analogue) as an antifungal drug requires the demonstration that human inositol polyphosphate kinases are not affected. How the authors would achieve this, is not discussed.
Response: Our immediate goal is to improve specificity for inhibition of fungal IP3-4K over human IP3-4K because fungal IP3-4K plays a bigger contribution to fungal virulence/ viability than fungal IP6K does. It is possible that the final derivatives will inhibit IP6 kinases in both pathogen and host. However, IP6K off-target effects in the human host may not be significant given that patients are treated for fungal infections for a short time as opposed to those on metabolic drugs who receive these medications over a lifetime.
Round 2
Reviewer 1 Report
Some minor concerns for Figure 5:
1, Please label the hinge region.
2, In panels C and D, better to maintain the consistent color of Mg as in other panels.
3, Panel E, the labeling of P111 is off.
4, In panel E, better to show atoms of residues for one of the proteins with different colors, such as oxygen as red, and nitrogen as blue.
5, Still believe that showing valence bonds is not a good idea:
a, P-OH or P=O bonds within ATP might be mixed or not distinctly identified, showing it might be misleading.
b, The orientation of side chain for D144 could be adopted in another way.
c, Valence bonds didn't add more information.
d, There are many ways to show the difference of the main chain and the side chain.
Author Response
Some minor concerns for Figure 5:
1, Please label the hinge region
Response: This is now labelled in panels A and B
2, In panels C and D, better to maintain the consistent color of Mg as in other panels.
Response: Mg is now in violet, consistent with other panels.
3, Panel E, the labeling of P111 is off.
Response: Labelling is now all visible.
4, In panel E, better to show atoms of residues for one of the proteins with different colors, such as oxygen as red, and nitrogen as blue.
Response: HsIPMK is now depicted as suggested.
5, Still believe that showing valence bonds is not a good idea:
a, P-OH or P=O bonds within ATP might be mixed or not distinctly identified, showing it might be misleading.
b, The orientation of side chain for D144 could be adopted in another way.
c, Valence bonds didn't add more information.
d, There are many ways to show the difference of the main chain and the side chain.
Response: Yellow dashed lines are now removed.